# Predictive Factors in Metastatic Melanoma Treated with Immune Checkpoint Inhibitors: From Clinical Practice to Future Perspective

**DOI:** 10.3390/cancers16010101

**Published:** 2023-12-24

**Authors:** Stefano Poletto, Luca Paruzzo, Alessandro Nepote, Daniela Caravelli, Dario Sangiolo, Fabrizio Carnevale-Schianca

**Affiliations:** 1Department of Oncology, University of Turin, AOU S. Luigi Gonzaga, 10043 Orbassano, Italy; alessandro.nepote@unito.it; 2Department of Oncology, University of Turin, 10124 Turin, Italy; luca.paruzzo@unito.it (L.P.); dario.sangiolo@unito.it (D.S.); 3Division of Hematology and Oncology, Perelman School of Medicine, University of Pennsylvania, Philadelphia, PA 19104, USA; 4Medical Oncology Division, Candiolo Cancer Institute, FPO-IRCCs, 10060 Candiolo, Italy; daniela.caravelli@ircc.it (D.C.); fabrizio.carnevale@ircc.it (F.C.-S.)

**Keywords:** melanoma, immunotherapy, immune checkpoint inhibitors, predictive factor, biomarker

## Abstract

**Simple Summary:**

Metastatic melanoma treatment has greatly changed in the last decade due to the introduction of target therapies and immune checkpoint inhibitors. The combination of immune checkpoint inhibitors led to an unprecedented median overall survival of 72 months, but still, an important portion of patients did not significantly benefit from this approach. In this scenario, the identification of predictive factors is mandatory to improve treatment choices in daily practice. In this review, we summarize the most updated data on trials evaluating predictive factors in metastatic melanoma patients treated with immune checkpoint inhibitors, providing information to support daily practice decisions, and at the same time, highlighting the most promising future perspectives.

**Abstract:**

The introduction of immunotherapy revolutionized the treatment landscape in metastatic melanoma. Despite the impressive results associated with immune checkpoint inhibitors (ICIs), only a portion of patients obtain a response to this treatment. In this scenario, the research of predictive factors is fundamental to identify patients who may have a response and to exclude patients with a low possibility to respond. These factors can be host-associated, immune system activation-related, and tumor-related. Patient-related factors can vary from data obtained by medical history (performance status, age, sex, body mass index, concomitant medications, and comorbidities) to analysis of the gut microbiome from fecal samples. Tumor-related factors can reflect tumor burden (metastatic sites, lactate dehydrogenase, C-reactive protein, and circulating tumor DNA) or can derive from the analysis of tumor samples (driver mutations, tumor-infiltrating lymphocytes, and myeloid cells). Biomarkers evaluating the immune system activation, such as IFN-gamma gene expression profile and analysis of circulating immune cell subsets, have emerged in recent years as significantly correlated with response to ICIs. In this manuscript, we critically reviewed the most updated literature data on the landscape of predictive factors in metastatic melanoma treated with ICIs. We focus on the principal limits and potentiality of different methods, shedding light on the more promising biomarkers.

## 1. Introduction

Until the early 2000s, metastatic melanoma was an incurable disease with a median overall survival (OS) of about 6 months [1]. In recent years, the introduction of targeted therapies and immune checkpoint inhibitors (ICIs) has drastically changed this scenario, significantly improving patient outcomes. The anti-Cytotoxic T-Lymphocyte Antigen 4 (CTLA-4) antibody Ipilimumab was the first ICI to demonstrate important efficacy results in patients with advanced melanoma. Ipilimumab showed an ORR of 10.9% with a median OS of about 10 months [2,3]. Subsequently, treatment with anti-Programmed Cell Death Protein 1 (PD-1) antibody monotherapy showed an improvement as compared with Ipilimumab with an overall response rate (ORR) of 42–45% with a median progression-free survival (PFS) of 4.6–8.4 months and a median OS around 3 years [4,5]. In the CheckMate-067 trial, the combination of the anti-PD-1 Nivolumab and Ipilimumab showed an ORR of 58%, leading to a median PFS of 11.5 months with an unprecedented median OS of 72.1 months [6]. Even if this trial was not formally designed to compare the combination therapy with Nivolumab alone, these results led to the approval of Nivolumab + Ipilimumab as the standard of care in the treatment of patients with metastatic melanoma. Recently, another combination of ICIs with Nivolumab and the anti-Lymphocyte Activation Gene 3 (LAG-3) antibody Relatlimab was introduced into the field, showing an ORR of 43.1% and a median PFS of 10.1 months. The reduced toxicity with similar outcomes shown with this treatment suggests a possible important role for anti-LAG-3-based combination in the treatment of melanoma patients, but data on OS are still immature [7,8]. In patients with BRAF-mutated melanoma, another evaluated option is represented by the association of ICIs and target therapy. In the phase III IMspire150 trial, adding the anti-Programmed Cell Death Protein–Ligand 1 (PD-L1) Atezolizumab to the combination of Vemurafenib + Cobimetinib led to a significantly increased median PFS (15.1 vs. 10.6 months), with a similar ORR, while in the phase III COMBI-I trial the increase in median PFS did not reach the statistical significance [9,10]. Additional follow-up of the IMspire150 trial showed that overall survival was not significantly improved with triplet therapy (29 vs. 25.8 months), questioning the possible role of this combination [11].

Despite these outstanding results, many shadow areas need to be addressed to further improve disease prognosis. Even if some patients could obtain long-term benefits when treated with ICIs, almost half of them do not respond (primary resistance) or eventually develop progression after an initial response (acquired resistance). Due to the numerous options available in clinical practice, the identification of predictive factors of response to immunotherapy is a critical need to tailor the treatment of metastatic melanoma patients. Patients with a low probability of obtaining benefits when treated with ICIs could be treated earlier with target therapies or different immunotherapeutic approaches other than ICIs. In contrast, in patients who can be candidates for an immunotherapy-first approach, it is crucial to define which patients could be treated with anti-PD-1 monotherapy with the same benefit but fewer adverse events and which patients need a combination treatment. Moreover, ideal biomarkers could help assess the decision between the different ICIs combinations available and to define patients that could benefit from a triplet combination therapy with BRAF inhibitors, MEK inhibitors, and anti-PD-(L)1. 

Predictive factors of response to immunotherapy can be divided into host-associated factors, immune system activation-correlated factors, and tumor-associated factors (Figure 1).

In this review, we critically summarize the most important factors described to be associated with response to ICIs, providing clinicians a comprehensive review of the most updated literature data, and, at the same time, highlighting the most promising future perspectives. For some of the described factors, data are still controversial and need further studies to permit their introduction into clinical practice, while for others, data are mature and could be useful in some way in clinical decision-making. For this reason, we present the most discussed factors schematically divided into three groups. In the first one, we reviewed predictive/prognostic factors with strong evidence that could potentially affect decision-making (Table 1). In the second one, we reviewed predictive/prognostic factors with less evidence and/or potentially less useful in decision-making (Table 2). In the third one, we reviewed predictive/prognostic factors with limited and/or controversial data and not affecting to-date clinical decision-making (Table 3).

## 2. First Line Predictive/Prognostic Factors

### 2.1. Antigen Presentation

A nodal point to restore an immune response is the neoantigen presentation, which, in this scenario, plays a key role for the molecules that form the Human Leukocyte Antigen (HLA) complex. One of the best-known mechanisms of resistance to ICIs is represented by the downregulation, alteration, or complete loss of the molecules of the HLA class-I complex, such as beta2-microglobulin [12,13]. Rodig et al. showed that the loss of HLA class-I expression on melanoma cells determined by immunohistochemistry was associated with resistance to anti-CTLA-4 but not to anti-PD-1 [14]. Another study showed that HLA class-I expression was intact in all patients with durable response to anti-PD-1 monotherapy but not in all patients with durable response to combination immunotherapy. Tumors with low expression of HLA class-I antigens displayed reduced T-cell infiltration and a myeloid-suppressor cells-enriched microenvironment. This finding suggests that combination immunotherapy could overcome an impaired antigen presentation and could be the preferred option in HLA class-I low-expressing melanoma [15]. The different effects of HLA class-I loss on response to anti-CTLA-4 and anti-PD-1 is not fully understood, but it could be explained by other immunological pathways, among which HLA class-II. A higher expression of HLA class-II molecules was found in responders as compared with non-responders in patients treated with anti-PD-1 monotherapy but not with anti-CTLA-4 [16]. Johnson et al. confirmed that HLA class-II expression on tumor cells identified by immunohistochemistry was associated with clinical response, PFS, and OS in melanoma patients treated with anti-PD-1, as well as with CD4+ and CD8+ tumor infiltrate [17]. However, the role of HLA class II is more multifaced. Engagement of HLA class-II molecules on melanoma cells by its ligand LAG-3 expressed on melanoma-infiltrating T cells might trigger an immune escape pathway [18]. Besides the loss of HLA expression, several studies associated a different expression in HLA class-I molecules with response to ICI and survival. Patients with different cancer types experienced longer OS if they had heterozygosis in all HLA class-I loci. This difference appears more evident when evaluating contemporaneously HLA heterozygosis and TMB [19]. Naranbhai et al. found that HLA-A*03 allele count was associated with poor OS and PFS in several different tumor types, but the mechanism of this effect was not described and remains unclear [20]. The complexity of this evaluation is complicated by the presence of numerous intracellular proteins that participate in the process of antigen formation and the exposure on the cell surface, the so-called antigen-presenting machinery (APM). A quantitative variation in APM components was proposed as a factor influencing survival in melanoma patients [21]. Thompson et al. generated a score based on the expression of eight APM molecules, evaluating two independent cohorts of melanoma and NSCLC patients. A higher APM score was associated with improved disease-free survival and OS, and a very low APM score was particularly powerful in identifying those patients who did not respond [22]. Even if the role of HLA as a predictive factor is established, some issues remain unanswered. Importantly, future studies will clarify which features better correlate with response to ICI and the best method for its evaluation. The role of APM is promising but should be explored in larger trials.

### 2.2. C-Reactive Protein (CRP)

CRP is an independent prognostic biomarker related to high tumor burden and associated with shorter OS and melanoma-specific survival [23,24]. The predictive role of CRP is irrespective of the type of ICIs. An analysis from the CheckMate-064, 066, and 067 trials highlighted that higher CRP at baseline negatively correlates with response rate and OS after Ipilimumab alone, Nivolumab alone, or their combination [25,26]. 

### 2.3. Circulating Tumor DNA (ctDNA) 

In the last decade, ctDNA has emerged as a promising predictive and prognostic factor. In 2014, Lipson et al. highlighted how high baseline levels of ctDNA correlated with worse outcomes in melanoma patients [27], while in 2020, Marsavela et al. showed that low pre-treatment plasma ctDNA levels were correlated with longer PFS in patients with melanoma receiving ICIs in the first line, but not in the second line [28]. Lee et al. confirmed that undetectable ctDNA at baseline is an independent predictor of prolonged survival in melanoma patients treated with anti-PD-1 with better performance compared to other parameters such as LDH, disease burden, and Eastern Cooperative Oncology Group (ECOG) performance status [29]. Thus, ctDNA is a surrogate for tumor burden and a prognostic biomarker for survival in metastatic melanoma patients.

However, the role of ctDNA is limited in patients with brain metastases. In the study of Seremet et al., patients who experienced disease progression exclusively in the central nervous system had undetectable ctDNA, probably due to the blood–brain barrier [30].

CtDNA could also be used as a non-invasive method to predict tumor evolution due to the possibility of performing periodic evaluations to assess dynamic changes in ctDNA levels during treatment [31]. An increase in ctDNA levels could anticipate the radiological progression assessed with a CT scan, possibly permitting an early interruption of ineffective treatment. This real-time evaluation could also be useful to distinguish cases of pseudo-progression from real progression [27,32,33]. However, limitations still exist due to the presence of false positive patients, as described in the study by Nabet et al. in non-small cell lung cancer patients where ctDNA early kinetics misclassified 25% of patients with durable clinical benefit [34]. Another issue is represented by the fact that 25% of melanomas do not present an identifiable molecular driver that could be monitored, leaving uncovered part of melanoma patients [29].

### 2.4. IFN-Gamma Gene Expression Profile (GEP)

With the spread of next-generation sequencing (NGS) techniques in the last decade, multiple novel biomarkers could be studied to predict responses to cancer treatment due to the possibility of studying cancer genomes. Multiple sequences of GEP have arisen in recent years, although they often lack external validation. 

One of the most validated GEPs is the IFN-gamma signature, referring to all the transcriptomics sequences that reflect T-cell activation and INF-gamma production. INF-gamma production is critical for innate and adaptive immune activation in melanoma but has a role also in differentiation as well as in tumor cell apoptosis and senescence [35]. This pathway is influenced by various factors, including tumor microenvironment composition features, such as TILs level, T-cell phenotype, and the presence of myeloid-derived suppressor cells [35]. High INF-gamma levels are related to a better overall response to immunotherapy in melanoma [16]. In preclinical studies, loss of the IFN-gamma signaling pathway was associated with primary resistance to anti-CTLA-4 therapy [36]. 

Multiple IFN-gamma GEPs have been proposed. Most of them include significant actors in INF-gamma machinery, such as CXCL-9 and HLA gene expression [37,38,39,40,41]. The role of IFN-gamma signature in the prediction of response to ICIs in melanoma patients was proved in several studies. Grasso et al. retrospectively analyzed 101 patients with melanoma treated with Nivolumab alone or combined with Ipilimumab, showing that the antitumor T cell response and the corresponding downstream IFN-gamma signaling are the main drivers of clinical response or resistance to ICI therapy [38]. Yan et al. proposed a 9-gene IFN-gamma signature, showing a higher score in patients who achieved complete or partial response to immunotherapy [39]. Ayers et al. evaluated a T cell-inflamed GEP containing IFN-gamma-responsive genes related to antigen presentation, chemokine expression, cytotoxic activity, and adaptive immune resistance. This model predicted response to pembrolizumab across multiple solid tumors, including melanoma. However, some patients with high IFN-gamma signature expression showed immune resistance, suggesting that an inflammatory microenvironment is necessary but not sufficient to achieve a response [37]. Tumoral intrinsic INF-gamma signaling has also shown in murine models an ambiguous pro-tumorigenic activity, contributing to melanoma progression by inhibiting cell apoptosis and enriching the tumor microenvironment of immunosuppressive polymorphonuclear leukocytes [42].

The most important evidence on IFN-gamma signature comes from prospective trials evaluating neoadjuvant therapy due to the possibility of testing response to therapy directly on tumor tissue from surgical specimens. In the Opacin-NEO trial, Rozeman et al. used an IFN-gamma GEP including all the genes described by Ayers et al., finding that 95% of patients with high IFN-gamma signature achieved a pathological response compared with 57% of patients with low IFN-gamma scores, with lower risk of relapse at 2 years (90% vs. 71%). Patients with high IFN-gamma scores and high tumor mutational burden (TMB) had 100% complete pathological response, while patients with low IFN-gamma scores had 88% and 39% complete pathological response, depending on high or low TMB. These data strongly suggest a potential role for IFN-gamma signature and the association of IFN-gamma signature and TMB as a valuable predictor biomarker [40]. Reijers et al. confirmed in a cohort derived from Opacin-NEO and PRADO trials that IFN-gamma score was significantly associated with response to neoadjuvant immunotherapy: the pathological response rate was 89% in patients with a high IFN-gamma score versus 49% in patients with low score [41]. 

These data highlight the possibility of de-escalating immunotherapy in patients with a high IFN-gamma score and emphasized the need for more effective treatment combinations for patients with a low IFN-gamma score. Ongoing studies will evaluate this intriguing possibility: in the DONIMI trial, patients with a high IFN-gamma score had higher pathologic response rates despite treatment de-escalation compared to patients with a low score [43].

All studies reported here analyzed IFN-gamma score or expression profile on tumor simple. As far as we know, no study has been conducted to indicate if IFN-gamma score expression on liquid biopsy may be a surrogate for tumor samples.

INF-gamma signatures are an important biomarker of response to immunotherapy, and their role was established especially in the neoadjuvant setting. The widespread of NGS techniques and their cost reduction will permit their introduction into daily routine, but further evidence is needed to permit prospective validation of IFN-gamma GEP and guarantee its reproducibility.

### 2.5. Lactate Dehydrogenase (LDH)

Elevated serum LDH is the mirror of a high tumor burden and an independent negative prognostic factor in metastatic melanoma [44,45]. Several studies showed that elevated LDH at baseline is associated with shorter OS during ICI therapy [45,46,47]. This can also be related to preclinical data in which LDH is a key regulator of glycolysis and a possible determinant of an immunosuppressive microenvironment [48,49]. 

### 2.6. Number and Location of Metastatic Sites

Stage group is a prognostic factor in metastatic melanoma [44], but the predictive value of tumor burden, described as number of metastatic sites, in patients treated with immunotherapy is not completely established. 

In a large pooled analysis based on patients treated in the pivotal trial with Dabrafenib + Trametinib, Long et al. found that baseline LDH level and number of metastatic sites were the most important factors in predicting PFS and OS [50]. To date, similar results in patients treated with ICI have not been published. Retrospective studies showed an association between a number of metastatic sites and prognosis in different solid tumors [51].

In a recent study based on 1644 patients treated with anti-PD-1 alone or in combination with Ipilimumab, Pires da Silva et al. found a clinical model to predict the ORR, PFS, and OS that did not comprise the number of metastatic sites but included the presence of lung and liver metastases [52]. This work highlights that differences in involved organs could influence the response to immunotherapy. A profound difference in the tumoral microenvironment in different metastatic sites explains these results: liver and brain metastases have less T-cell infiltration as compared with other sites. In particular, liver metastases present fewer PD-1+ T-cells but a higher presence of TIM-3+ T-cells [53]. Similarly, Tumeh et al. observed that the presence of liver metastasis was associated with reduced response as compared with patients without liver metastasis. A liver biopsy revealed a reduced CD8+ T-cell density at the invasive tumor margin, confirming an immune desert microenvironment in liver metastases [54].

The role of other metastatic sites frequently present in melanoma, such as bone metastases, is less established. Further studies are needed to clarify the peculiarity of each metastatic site. 

### 2.7. Tumor Mutational Burden

TMB, defined as the mutational load carried by the tumor per megabase, has been extensively investigated in the last decade [55,56].

Based on evidence from the KEYNOTE-158 study, in 2020 FDA approved Pembrolizumab for patients with a TMB ≥ 10 mutation/Megabase, irrespective of tumor histology. However, several questions about TMB as a predictive biomarker remain unanswered, such as the right threshold in different cancer types [55,57].

In melanoma, the real utility of TMB as a predictive biomarker is still controversial. In an exploratory analysis from CheckMate-066 and CheckMate-067 trials, high baseline TMB was associated with improved treatment efficacy in terms of ORR, PFS, and OS in patients receiving Nivolumab alone or in combination with Ipilimumab or Ipilimumab alone, especially in BRAF wild-type patients. Median TMB was similar for responders and non-responders in BRAF-mutant patients, whereas TMB was higher for responders in BRAF wild-type. The performance of TMB was increased when associated with an inflammatory signature, suggesting the potential predictive value of a combined approach [58].

In the OpACIN-neo trial, high TMB correlates with better response irrespective of IFN-gamma signature, although only the association had the statistical power to identify patients with complete pathological response [40]. Analysis from the IMspire150 and COMBI-I trials focused on a possible role of TMB to predict an advantage of adding ICI to target therapy, but baseline LDH level remains the main factor in determining response to treatment [59,60].

Liu et al. showed that TMB cannot be applied generically across melanoma subtypes. In 144 melanoma patients, TMB was higher in responders than in progressors, but this difference was not confirmed stratifying for melanoma subtypes. TMB was significantly higher in cutaneous and occult melanomas as compared with acral and mucosal. Interestingly, responders with mucosal or acral melanoma had a lower TMB than progressors with cutaneous or occult melanoma, suggesting that the disease subtype confounds the association between TMB and response to ICI [16].

In conclusion, even if the potential of TMB as a predictive biomarker is clear, its role is not completely established in melanoma. TMB can be correlated with other easily evaluable predictive factors, such as melanoma subtype or mutational status. Differently from other cancer types, the role of TMB in melanoma is probably part of a more complex model integrating clinical and genomics features.

## 3. Second Line Predictive/Prognostic Factors

### 3.1. Checkpoint Immunohistochemical Evaluation

The rationale of every ICI is to limit the contact of inhibitory ligands on the tumor membrane with their counterpart on T cells. Starting from that point, the immunohistochemistry quantification of these checkpoints has been evaluated in multiple cancer types as a predictor of response to immunotherapy. In the melanoma setting, the most evaluated checkpoints are PD-L1 and LAG-3. 

PD-L1 is a transmembrane protein expressed in normal immune cells as well as in tumor cells. In the tumor, this molecule suppresses immune surveillance and promotes tumor growth [61]. PD-L1 expression on tumor cells could also be upregulated by IFN-gamma [62]. Immunohistochemical (IHC) expression of PD-L1 has been evaluated in pivotal clinical trials with anti-PD-1 therapies as a predictor of response and outcome. In the CheckMate-066 and in the KEYNOTE-006, anti-PD-1 monotherapy (Nivolumab and Pembrolizumab, respectively) improved clinical outcomes as compared with the standard arm independently from PD-L1 expression. In the Checkmate-066 ORR was numerically superior in the PD-L1+ population (52.7% vs. 33.1%) [4,63]. In the CheckMate-067 trial, both Nivolumab and the combination of Nivolumab + Ipilimumab performed better than the comparator arm Ipilimumab, but the benefit of the combination therapy was more evident in the PD-L1 negative group (median event-free survival 11.2 vs. 5.3 months) than in patients with PD-L1 expression more than 1%, in which there were not differences in terms of clinical outcomes [6]. However, these results must be interpreted with caution as they derive from a non-prespecified analysis. A German study showed that PD-L1 expression was different based on the anatomic site of the biopsy, and only the expression in certain tissue types has some correlations with outcomes [64]. These data globally confirm that PD-L1 is not a good predictor of response to anti-PD-1 in melanoma, but it may have a role in identifying patients as candidates for an immunotherapeutic combination approach. 

LAG-3 is a novel inhibitory checkpoint that negatively regulates proliferation and effector function in T cells. The LAG-3 and PD-1 pathways are distinct inhibitory immune checkpoints that are often co-expressed on tumor-infiltrating lymphocytes. Thus, the interaction with their ligands contributes to tumor-mediated T-cell exhaustion [65].

In the phase III Relativity-047 trial, the combination Nivolumab + Relatlimab showed improved ORR and median PFS over Nivolumab alone, independently from LAG-3 and PD-L1 expression [7]. However, a recent study showed a significantly higher proportion of LAG-3+ cells in responders to Nivolumab plus Relatlimab as compared to non-responders. Also, patients with a higher expression of LAG-3+ cells in their tumors had significantly longer PFS, but no significant difference was observed in OS. These data suggest that the assessment of LAG-3 expression via IHC warrants further evaluation to determine its role as a predictive marker of response and survival in metastatic melanoma [66]. However, in our opinion, the immunohistochemistry evaluation of these markers can be compromised by many concomitant technical and immune microenvironmental factors. Therefore, these data must be integrated with the information derived from the tumor immune cells. 

### 3.2. Gut Microbiome

The gut microbiome was found to influence the interplay between host and cancer, modulating response to immunotherapy. Patients with a favorable microbiota profile show higher levels of CD4+ and CD8+ T cells, whereas patients with an unfavorable profile have higher levels of T regulatory cells (Treg) and Myeloid-derived suppressor cells (MDSC), suggesting a role in the development of hotter tumor microenvironment [67,68]. Microbiota could induce Interferon (IFN) production by intratumoral monocytes, which trigger the recruitment and activation of natural killer (NK) cells [69]. Some effects could also be mediated by the production of short-chain fatty acids [70].

However, an ongoing debate exists on which microbiome characteristics are associated with treatment response. In the first study on metastatic melanoma patients in 2017, Frankel et al. found that *Bacteroides caccae* was enriched in all ICI responders, while *Fecalibacterium prausnitzii*, *Bacteroides thetaiotamicron*, and *Holdemania filiformis* were higher in responders to ICI combination and *Dorea formicogenerans* was enriched in anti-PD-1 responders [71]. Gopalakrishnan et al., in the same year, found in a larger study an enrichment of *Ruminococcaceae*, *Clostridiales*, and *Faecalibacterium* in responders, whereas non-responders presented a dominance of *Bacteroidales* [68]. Chaput et al. found that patients with a higher proportion of *Bacteroides* and other *Firmicutes* had reduced PFS and OS after treatment with Ipilimumab, while *Faecalibacterium* was higher in patients with long-term benefits [67]. In the study of Matson et al., eight species were more abundant in responders, whereas *Ruminococcus obeum* and *Roseburia intestinalis* were more abundant in non-responders [72]. Peters et al. associated *Coprococcus eutactus*, *Prevotella stercorea*, *Streptococcus sanguinis*, *Streptococcus anginosus,* and *Lachnospiraceae bacterium* with longer PFS, whereas *Bacteroides ovatus*, *Bacteroides dorei*, *Bacteroides massiliensis*, *Ruminococcus gnavus*, and *Blautia products* were present in patients with poor clinical outcomes [73].

Therefore, various bacteria were associated with response to ICIs, and nowadays, there is a lack of consensus. A meta-analysis comprising these studies tried to summarize these results. Interestingly authors found that associations between outcomes and microbial signatures are more robust for unfavorable taxa than favorable taxa. This work also shows that microbiota influence on response becomes a dominant factor at 9–10 months after initiation of treatment, suggesting that favorable microbiota need to be maintained to prevent disease progression [74]. More recently, an Australian-Dutch study confirmed an association between the presence of Ruminococcaceae or Bacteroidaceae and clinical outcomes [75].

Probably the role of gut microbiome is more complex than previously thought and not dependent on a single species. Diverse microbiota compositions, as well as several microbially derived products, may likely result in the same downstream immunomodulatory effect. For example, microbiota species with specialized peptidoglycan remodeling activity and muropeptide-based therapeutics were demonstrated to enhance immunotherapy [76].

In addition, not only gut microbiota can impact response to ICIs. Patients with serology positive for Helicobacter pylori showed reduced ORR, PFS, and OS [77]. 

Microbiota is influenced by several factors, such as sex, age, geography, drugs, diet, and nutritional status. Therefore, this field is not completely explored but his understanding is required not only to implement this non-invasive biomarker assessment but also for the important therapeutic implications of manipulating the microbiome, such as via fecal transplantation or diet modification. A recent study showed that responder-derived fecal microbiota transplantation can change the gut microbiome and reprogram the tumor microenvironment to overcome resistance to anti-PD-1 [78,79,80].

### 3.3. IL-6

Different cytokines and soluble serological markers have been associated with ICI response. The pro-inflammatory interleukin-6 (IL-6) has been associated with a poor response and shorter survival in patients receiving single-agent ICIs [25,26,81]. Moreover, in CheckMate-064, a decrease in IL-6 level at 3 months was associated with longer OS compared to patients who do not experience this decrease [26].

### 3.4. Immune-Related Adverse Events (irAEs)

With the introduction of immunotherapies as the standard of care for many malignancies, clinicians have learned to deal with a new spectrum of toxicities. The development of irAEs is due to a non-specific reactivation of the immune system [82]. IrAEs may frequently involve different organs, such as the skin, endocrine system, lung, gastrointestinal tract, and liver. Other toxicities, such as cardiovascular and neurological disease, are less frequent but could lead to a high rate of mortality and comorbidity [83,84]. The combination of Nivolumab and Ipilimumab was associated with a higher incidence of irAEs [85]. However, this combination led to a longer treatment-free interval in patients who discontinue treatment due to AEs [6].

Several studies correlated the occurrence of irAEs with a benefit in terms of ORR, PFS, and OS [86]. The clearest evidence of this phenomenon came from melanoma patients who developed vitiligo-like depigmentation after different immunotherapy regimens. Vitiligo development was associated with an improved PFS and OS, with a two to four times less risk of disease progression and death compared to patients who did not develop vitiligo [87]. A meta-analysis by Xing et al. investigated the correlation between the occurrence of various irAEs and response in advanced solid tumor patients treated with Nivolumab or Nivolumab + Ipilimumab [88]. The ORR in patients treated with Nivolumab was positively correlated with the occurrence of skin, gastrointestinal, and endocrine toxicities, while there was a negative correlation with pulmonary irAEs. In patients treated with Nivolumab + Ipilimumab, the positive correlation was maintained only for skin and gastrointestinal toxicities. Interestingly, when melanoma patients were excluded from the analysis, the negative correlation between ORR and pulmonary toxicities in the Nivolumab arm was not observed. These results were confirmed in a Japanese cohort including 60 melanoma patients treated with Ipilimumab after progression on anti-PD-1. In a multivariate analysis, the occurrence of skin or endocrine toxicities was correlated with better overall survival [89]. 

These data suggest that the experience of an irAE may be a predictor of response to ICIs in melanoma. Evidence is stronger for some types of toxicities (e.g., skin), while the role of different organ involvement is less proven. For this reason, the meaning of different grades and types of toxicity needs to be further explored.

### 3.5. Radiologic Assessment and Radiomic

CT scan is the standard of care for pre-treatment evaluation and tumor response assessment. However, a 2-fluoro-2-deoxy-D-glucose-positron emission tomography (FDG-PET) scan can capture the metabolic activity of tumor lesions to synergistically assess treatment response. A post hoc analysis from the KEYNOTE-001 and KEYNOTE-006 trials showed that an important portion of patients with an initial stable disease (SD) could achieve subsequent partial or complete responses with similar outcomes as compared with patients with early partial or complete response [90]. However, it could be difficult to predict which patients with initial SD could eventually achieve an objective response. For this reason, management of patients with early SD is an important clinical challenge and PET-FDG could help clinician decisions. Several studies showed that PET-FDG could better predict long-term outcomes in melanoma patients treated with anti-PD-1 [91,92]. In contrast, the predictive role of FDG-PET at baseline, as well as its interaction with baseline disease characteristics, should still be examined in further studies. In a report by Seban and colleagues based on 55 melanoma patients treated with anti-PD-1, a high metabolic tumor volume, an increased spleen-to-liver uptake, and bone marrow-to-liver uptake ratio were predictors of survival in multivariate analysis [93]. In the era of machine learning and artificial intelligence, a promising novel approach is represented by radiomics, a multi-step analysis based on the extraction of various tumor-specific features directly from routine radiological images. Tumor characteristics evaluated with radiomics, such as tumor volume, morphology, and density, could better reflect tumor biology and the entire tumor burden. Trebeschi et al. found that lesions with more heterogeneous morphological profiles with non-uniform density patterns and compact borders were more likely to respond to immunotherapy [94]. Other studies correlated outcomes with different radiomic features, such as kurtosis or skewness [95,96]. Kurtosis is a feature associated with vascularization in the tumor microenvironment, and authors hypothesized that this may reflect a higher T-cell infiltration within the lesion [97]. In trying to obtain a more precise estimation of the outcome, different radiomic features could be combined to create a radiomic signature. Dercle et al. found that a radiomic signature that combines four imaging features (two related to tumor size and two reflecting changes in tumor imaging phenotype) could predict prognosis in melanoma patients treated in two Pembrolizumab trials [98]. Sun et al. validated a radiomic signature that predicts the CD8+ T-cells tumor infiltration, providing a non-invasive tool to estimate tumor phenotype. They found that this signature could predict outcomes at a lesion level in advanced melanoma patients treated with anti-PD-1. In addition, analysis of intralesional heterogeneity may help to identify lesions at the highest risk of progression with a direct impact on the patient prognosis. Future studies should define if a locoregional approach direct to a lesion at a higher risk of progression could improve a patient’s prognosis [99,100]. Radiomic signatures could also help to identify lesions with a high risk of hyperprogression as well as to predict overall survival in patients with brain metastases [101,102]. Even if the potential advantage of radiomic is clear, several limitations to the daily application remain. Radiomics can depict the entire tumor load, allowing all lesions to be analyzed, but the time necessary for the manual segmentation of tumor lesions is a limit for routine application. Furthermore, much data come from single-center retrospective studies: the use of different imaging techniques, as well as different software and feature selection, limit the reproducibility of these models. Multicentre prospective studies are needed to confirm the prognostic value of radiomics and to incorporate it into clinical practice.

### 3.6. Complete Blood Count

Recent studies have shown that an elevated baseline neutrophil-to-lymphocyte ratio (NLR) is negatively associated with OS and PFS during anti-CTLA-4 and/or anti-PD-1 treatment [103]. However, not all patients with higher-than-normal NLR do not respond to ICIs. Multiple studies identify a ratio of 5 as a cut-off able to predict response to immunotherapy [104], but large clinical trials should confirm this cut-off to permit its use in daily routine. Moreover, this parameter can be influenced by concomitant medications (e.g., steroids) or other host factors (e.g., infection), representing an epiphenomenon of a worse performance status. For that reason, NLR should be integrated with a holistic vision of the patients. For example, in a recent study, Pires da Silva et al. used a multiparametric nomogram that include NLR to successfully predict PFS and OS in metastatic melanoma patients treated with ICIs [52].

The complete blood count at baseline could provide more information reflecting the immunological and inflammatory status of the patients. Other than the well-studied NLR, the role of the lymphocyte-to-monocyte ratio (LMR) and platelet-to-lymphocyte ratio (PLR) has been evaluated as ready-to-use tools. In a retrospective cohort, Lobo Martins et al. showed that high PLR and LMR were negatively correlated with PFS, but only PLR maintains its significance in multivariate analysis. The role of these markers is limited since no large clinical trials have confirmed these results [105].

### 3.7. Peripheral Blood Mononuclear Cell Populations

Peripheral Blood Mononuclear Cell (PBMC) subsets in the peripheral blood are associated with response to immunotherapy. Patients that respond to anti-PD-1 present lower levels of CD8, CD4, γδ T cells but higher levels of NK and CD19^−^HLA-DR^+^ myeloid cells. Within the T-cell compartment, the peripheral phenotypes have been reported as associated with different clinical outcomes. In the metastatic melanoma setting, baseline levels of CD45RO8/CD8+ T cells and CD8+ effector memory predict response to immunotherapy, while the abundance of PD-1+CD56+ T cells inversely correlates with PFS and OS. On the other hand, within the myeloid compartment, the frequency of classical CD14^+^CD16^−^CD33^+^HLA^−^DR^hi^ monocytes might serve as a potential predictive factor for PD-1 blockade immunotherapy [106].

### 3.8. Tumor Microenvironment

The intra-tumor immune composition is a crucial factor influencing response and survival to ICIs [107]. Melanoma lesions present higher infiltration of T cells and lower myeloid cells, as compared to historically “cold” tumors such as pancreatic adenocarcinoma [108]. The evaluation of these two components could be of interest as a promising predictive factor of response to ICIs.

#### 3.8.1. Tumor-Infiltrating Lymphocytes

The quantitative evaluation of tumor-infiltrating lymphocytes (TILs) in formalin-fixed tumor biopsy ranges from visual scoring to multiplex immunohistochemistry, followed by enumeration using software analysis. However, TILs evaluation, at the moment, relies upon single marker immunohistochemistry analyses of major lymphocyte subsets [109].

Li et al. showed in a meta-analysis that higher infiltration of CD8+ cells correlates with improved ORR and PFS in cancer patients treated with ICIs [110]. However, in the era of single-cell tumor evaluation, deeper characterization of the T-cell populations has become mandatory. In a recent report, RNA single-cell analysis of melanoma metastatic lesions led to the identification of TCF7+CD8+ T cells as a predictor of response to immunotherapy (anti-CTLA-4, anti-PD-1, or their combination), while higher expression of exhausted CD8+ T cell (LAG-3+ and TIM-3+) has been associated with ICI failure [111]. Moreover, not only the subset of T infiltrating cells but also the topographical site within the tumor seems to be a predictor of response to immunotherapy: stromal or invasive CD8+ T cells correlate with better clinical outcomes in immunotherapy-treated cancer patients [112]. The next generation of multiplexed imaging, like the CODEX or MERFISH, will permit simultaneous evaluation of the T cell subsets and their spatial distribution, deeply defining the tumor microenvironment (TME) and novel ways to overcome the immunosuppressive tumoral milieu.

#### 3.8.2. Myeloid Cells

Considering the crucial role of myeloid cells as modulators of T effector cell function in cancers, their presence in the TME can influence response to ICIs [113].

Jiang et al. showed that gene signatures of tumor-associated macrophages (TAMs) and MDSC in melanoma samples were associated with reduced T-cell infiltration and resistance to anti-PD-1 and anti-CTLA-4 therapy [114]. Hugo et al. showed that pre-treatment biopsies enriched in CD163+ macrophages are associated with resistance to immunotherapy [115].

Even if myeloid tumor infiltration seems to be globally associated with a negative outcome, scRNAseq on treatment biopsy showed that TAMs of responder patients have expression signatures based on CXCL10 and CXCL11, identifying a possible subset of myeloid cells that can have a favorable role in response to ICI [116].

Another important aspect that should be considered when analyzing the TME is the spatial distribution of the different cell types and their possible interaction. On a small cohort of anti-PD-1-treated melanoma patients, non-responders showed higher proximity of CD68+ cells to CD8+ T cells as compared to responders, both in pre-treatment and on-treatment biopsies pointing out that the interaction of the myeloid cells with the T cell determines the immunosuppressive role of this type of cells [117].

Considering these data, also for this subset of immune cells, high-throughput methods that allow single-cell evaluation with detailed topographical evaluation will redefine the role of myeloid cells in predicting response to immunotherapy.

## 4. Third Line Predictive/Prognostic Factors

### 4.1. Age

Elderly patients represent an intriguing setting for cancer immunotherapy due to their immune system impairment. This immune senescence is characterized by a reduction in effector T-cells [118] and an increase in T-regulatory cells and myeloid suppressor cells [119]. Also, pharmacokinetic and microbiome differences could lead to different outcomes [120].

In pivotal clinical trials, older patients were usually underrepresented. However, several real-world data and meta-analyses showed similar efficacy and toxicities as compared to the younger ones both with ICI monotherapy and combination [121,122,123,124,125,126,127,128,129]. Discontinuation appears more frequent [130,131], probably reflecting the different impacts on quality of life due to similar adverse events (AEs) [132]. Some studies showed a trend favoring older patients, but the difference was not statistically significant [133,134].

Therefore, the role of aging on the immune system and ICI response should be better clarified. Age should not be considered for treatment decisions, but a more comprehensive frailty assessment (e.g., G8 screening) should be taken into consideration in older patients to identify a subgroup with a higher risk of AEs and hospitalization [135,136].

### 4.2. Body Mass Index (BMI)

Obesity is an established risk factor for many tumors but seems to confer a survival advantage, especially in patients treated with immunotherapy, a phenomenon known as the obesity paradox [137,138].

In preclinical models, obesity-associated chronic inflammation may generate an immune-suppressive microenvironment via the induction of dysfunctional T cells that can be reverted by treatment with ICIs [137]. Also, obesity is associated with metabolically quiescent tumors, with the downregulation of oxidative phosphorylation and multiple other metabolic pathways [139]. Finally, obesity could be linked to the gut microbiome [140].

In melanoma patients, this association was first seen by McQuade et al. in a large study that retrospectively analyzed six independent cohorts with a total of 2046 metastatic melanoma patients treated with targeted therapy, immunotherapy, or chemotherapy. In this study, obesity was associated with improved PFS and OS in targeted therapy and immunotherapy cohorts but not in patients treated with chemotherapy. However, the benefit was limited to male patients. In the cohort of patients treated with anti-PD-1, median PFS (7.6 vs. 2.7 months) and OS (26.9 vs. 14.3 months) were longer in obese men than in patients with normal BMI, but they showed no differences in women [141].

Successive studies showed similar results. A retrospective analysis of a small cohort of 66 patients treated with Ipilimumab showed significantly higher response rates in patients with a BMI above normal, with no statistically significant difference in PFS and OS [142]. Another retrospective study conducted on 130 patients treated in the first line with target therapy or immunotherapy showed that patients with an elevated BMI had an increased probability of achieving an objective response [143]. These results were confirmed in a meta-analysis of individual patient data on 3768 patients with multiple cancer types treated with ICIs: median OS was significantly longer in overweight/obese (20.7 vs. 11.3 months) with similar results observed in median PFS (8.3 vs. 3.7 months), also in the melanoma subgroup [144]. In a retrospective study of 1070 patients with advanced cancer treated with anti-PD-(L)1, Cortellini et al. found that obesity was associated with better PFS and OS and with an increased risk of immune-related AEs, confirming a possible interplay between weight and immune system [145,146].

Other studies showed contrasting results. In a prospective study on 423 melanoma patients, Donnelly et al. showed no difference in PFS or OS in the entire cohort, with a positive association in patients treated with a combination of anti-CTLA-4 and anti-PD-1, but not with anti-CTLA-4 or anti-PD-1 alone [147]. Similarly, a retrospective French study on 1214 melanoma patients found no association between BMI and PFS or ORR [148].

The inconsistency of the described study could be explained by different reasons. First of all, these studies were heterogeneous in terms of treatment schedules, study populations, and different statistical designs. In addition, they do not consider possible differences in pharmacokinetics between obese and normal-weight patients. Finally, BMI is not a perfect summary of body composition. For this reason, more recent studies have tried to use features of body composition to better explain this phenomenon.

A retrospective analysis that included 411 melanoma patients showed no association between OS and BMI, but weight loss was associated with shorter survival [149]. Another retrospective study performed on 139 melanoma patients showed that overweight/mild obese patients had a lower risk of progression and mortality compared to normal weight. This association was predominantly driven by males, who showed higher serum creatinine levels, a possible surrogate for skeletal muscle mass. These observations suggest that sarcopenia or direct measures of body mass composition may be a more suitable predictor of survival [150].

In the largest study that investigated the role of body composition in 287 patients with melanoma treated with ICIs, Young et al. did not observe associations between outcomes and BMI, but they described a trend toward worse outcomes in patients with sarcopenic obesity and lower muscle quantity and quality. This association was modest and not statistically significant. Authors conclude that body composition will likely not play a major role in clinical decision-making [151]. A more recent study used a CT-based body composition assessment. In the 107 analyzed patients, Faron et al. showed an advantage in overall survival in patients with higher amounts of skeletal muscle mass [152].

In conclusion, the role of BMI as a predictive factor is controversial. The described studies explain that more complex analyses of body composition could perform better since they can take into account a larger number of variables. Not only adipose tissue but also skeletal muscle could play a role in the function of the immune system [153]. Future studies should better define the role and the possible application of these features.

### 4.3. Concomitant Medications

In recent years, several medications have been supposed to influence the efficacy of immunotherapy via different mechanisms. However, even if a biological plausibility exists to justify an impact on response to ICIs, clinical data are largely derived from retrospective or post hoc analyses that include a few patients and different cancer types. Therefore, a debate is open whether concomitant medication could in some way affect response to immunotherapy or if there is just an associative relationship.

Given their immunosuppressive action, steroids were the first class of medications being investigated in this setting. Several studies found a negative correlation for baseline steroids [154,155]. However, this association could be affected by bias, as steroids are used to treat different cancer symptoms (e.g., dyspnoea, pain, and cerebral edema). The negative effect on outcome was confirmed in patients taking steroids for supportive care [156,157]. In contrast, the role of steroids used to mitigate adverse events is more controvert since studies showed different results, but a possible role was supposed, especially when high-dose steroids were used early during treatment [158,159,160,161]. Thus, steroids should be used with caution. The effect of different immunosuppressive drugs on immunotherapy should be further explored [162,163].

Special attention was even paid to the use of antibiotics. Different studies, including systematic reviews and meta-analyses, suggest that antibiotic exposure before the start of treatment with ICIs may negatively affect outcomes, probably inducing dysbiosis [164,165,166]. However, antibiotics use could reflect a status of worse performance status and poor physical and immunological conditions rendering it a surrogate marker of the dismal outcome, independently of its effects on the gut microbiome and ICI efficacy. In some cohorts, antibiotic use was not a predictive factor of response to ICI [167,168]. Interestingly, not only exposure to antibiotics themselves but also the timing and class of antibiotics could have a different impact [169,170]. However, the complexity of these interactions is far from being fully understood.

In the same way, proton pump inhibitors (PPIs) are frequently prescribed and could modify the microbiome [171]. However, most studies, including two meta-analyses, do not confirm their impact on the outcome, especially in melanoma patients treated with ICI monotherapy or combination [172,173,174,175]. The effect of PPIs could be cumulative, but the retrospective nature of these studies does not permit the assessment of the duration of PPI exposure. On the other hand, the association between PPIs and the comorbid condition is less strong than for steroids or antibiotics, confirming the hypothesis of an associative connection.

In the study of Buti et al. evaluating at the same time several medications, the authors found that only the three above-mentioned medications (steroids, antibiotics, and PPIs) were negatively associated with ORR, PFS, and OS [176]. Other drug classes are less studied in this context. The study by Cortellini et al. confirmed the association between baseline steroids administered for cancer-related indications, prophylactic antibiotics, and PPIs with worse clinical outcomes in patients treated with PD-1/PD-L1 checkpoint inhibitors. This study also shows that baseline statins, aspirin, and beta-blockers were correlated to an increased ORR, while anticoagulants and opioids were associated with worse PFS and OS. In contrast, NSAIDs and various anti-hypertensive drugs do not influence the outcome [177]. Wang et al. found no conclusive association between NSAIDs, beta-blockers, and metformin [178]. Recently, new studies investigated the possible role of glucose-lowering medications: findings were contrasting, and the effect of this class of drugs needs further studies [179,180]. Thus, with the more extensive use of ICIs, even in patients who assume polypharmacotherapy, the role of concomitant medication has to be further investigated in dedicated prospective trials.

### 4.4. Driver Mutations

Activating mutations in the oncogenes *BRAF* and *NRAS* are the most studied mutations in malignant melanoma. Genetic alterations in these genes can be found in approximately 40% and 20% of cases, respectively [181,182].

In BRAF-mutated melanoma, the choice of first-line treatment is based on clinical parameters. As suggested by several guidelines, patients for whom immunotherapy can be delivered safely for the first few months should be considered for immunotherapy first, while target therapy must be preferred in patients who need a prompt response to treatment [181]. Phase III DREAMseq recently showed an improved OS at 2 years with first-line Nivolumab + Ipilimumab as compared with first-line Encorafenib + Binimetinib (72% vs. 52%). Also, Nivolumab + Ipilimumab led to a better ORR in the first line than in the second line (46% vs. 30%), and this result was confirmed in the phase II SECOMBIT trial [183,184].

Some data also show that BRAF mutational status can influence the performance of ICI treatment. In the KEYNOTE-006, Pembrolizumab led to a similar response rate in BRAF wild-type and BRAF-mutant patients (41% and 43%) [185]. In the CheckMate-067 the combination of Nivolumab and Ipilimumab led to a 57% OS rate at 6.5 years and a not reached median OS in the BRAF-mutant population. In the subgroup analysis, the median PFS was 16.8 and 11.2 months in BRAF-mutant and BRAF wild-type patients, respectively. Moreover, the benefit of adding Ipilimumab to Nivolumab seems to be greater in the BRAF-mutant subgroup: PFS was 38% vs. 23% at 6.5 years in BRAF-mutant patients, while BRAF wild-type patients showed similar PFS in the two arms (34% vs. 31% at 6.5 years). However, CheckMate-067 was not designed to formally compare these treatment subgroups [6].

The presence of NRAS mutation is a controversial prognostic factor in metastatic melanoma, as different studies showed opposite results [186,187,188]. Even if trials evaluating new molecules are ongoing [189], to date, no effective target therapy exists for NRAS-mutant patients. Therefore, treatment for metastatic NRAS-mutant melanoma is based on ICIs. In recent work, Rose et al. showed in the multivariable analyses that treatment with anti-PD-1 + anti-CTLA-4 was associated with significantly improved PFS and OS as compared with anti-PD-1 alone, while these data were not confirmed in the NRAS/BRAF wild-type subgroup [190].

Even if these data suggest that a combination of ICI should be the preferred approach, only a large clinical trial that will use the NRAS mutation status to define treatment strategy can unveil the role of this marker in the setting of immunotherapy.

### 4.5. Performance Status

Patients with poor performance status (PS) were usually excluded from pivotal clinical trials, and the efficacy of immune checkpoint inhibitors (ICIs) in this population frequently comes from real-world data. Patients with poor PS have a worse outcome in terms of ORR, PFS, and OS. However, they experienced a similar safety profile, and an important portion of patients with poor PS could benefit from immunotherapy [52,191,192,193,194,195,196].

The inclusion of patients with poor PS in clinical trials should be supported to better understand the real role of ICIs in this population. The decision to treat ICI patients with poor PS should be well balanced, evaluating other factors that could affect treatment response, such as comorbidities and tumor burden.

### 4.6. Pre-Existing Conditions

With the spreading use of ICIs, an increasing number of treated patients present various comorbidities. Patients with pre-existing autoimmune disease (AID) raised concern for the possibility of flare-up of their AID or increased rate of other AEs. For this reason, these patients were frequently excluded from trials with ICIs. Due to the widespread use of ICIs in real life, several meta-analyses and retrospective analyses evaluated outcomes and safety in this population. Several studies showed a risk of exacerbating the pre-existing condition as well as an increased risk of developing different AEs [197,198,199,200,201,202]. The risk of flare appeared higher in patients with active disease as compared with patients with latent disease [203]. However, almost all these studies showed that AEs were easily manageable [199,204,205]. Lee et al. also reported an increased risk of cardiovascular events in patients with autoimmune disease [206]. However, in some studies, authors did not report a difference in AE rates [207]. A major limit of these studies is that they evaluated many AID in aggregate, and only a few focused on a specific AID. A meta-analysis involving 191 patients with psoriasis showed a 45% risk of flare-up, while efficacy was comparable to those of the general population [208,209]. In a different retrospective cohort, PFS was significantly longer in patients who experienced psoriasis flare versus those without (39 vs. 8.8 months, *p* = 0.049) [209]. Several other studies evaluated the possible influence of autoimmune disease on efficacy because some concerns exist about the possibility that immunosuppressive treatment for AID or pre-existing immune activation can decrease the efficacy of ICIs. Tang et al. evaluated 17,497 patients with pre-existing AID and 17,497 matched controls via the TriNetX Diamond network: patients with a history of AID were not at higher risk of mortality, while history of Hashimoto disease and vitiligo were statistically significantly associated with decreased mortality [210]. A French case–control study showed a better survival at 24 months in patients with pre-existing AID [202]. A report from the Memorial Sloan Kettering Cancer Center reveals that patients with AID who experienced irAEs had higher ORR (42.5% vs. 8.3%, *p* < 0.001) and significantly improved OS. Although patients with AID had similar response rates, they experienced significantly shorter OS [203]. A retrospective analysis based on 1781 patients showed that patients with abnormal baseline thyroid stimulating hormone (TSH) had inferior median OS (16 vs. 27 months; *p* < 0.001) [211]. A systematic review including 123 patients showed an ORR of 50% in patients who developed irAEs (either flare or other immune-related irAE), while in patients without any irAE, the ORR was 35.7%. Authors concluded that a flare of pre-existing AID or irAEs development may be associated with response to ICI [212]. In the Italian study of Cortellini et al., AID was not an independent factor influencing PFS or OS [213].

These data suggest that ICIs can be safely used in patients with AID, while there is no clear correlation with outcome. A special population is represented by patients with multiple sclerosis. A retrospective analysis of 11 patients treated at the Gustave Roussy Institute showed that only 9% of patients had a flare of multiple sclerosis, while no data on correlation with efficacy exists [214].

Since HIV could lead to a less effective immune system, there have been safety and efficacy concerns to include people living with HIV in clinical trials using ICIs. A recent retrospective analysis included 390 patients with HIV treated with anti-PD-(L)1-based therapies for advanced cancer, comprising 26 patients with melanoma [215]. In this population, authors observed an ORR of 47% with a median PFS of 5.9 months. In the same study, authors compared matched patients with and without HIV with metastatic non-small cell lung cancer, demonstrating similar outcomes and toxicities. This result supports the use of ICIs in patients living with HIV, highlighting the need for larger prospective trials.

Patients affected by other relevant comorbidities, such as cardiovascular disease, previous infections, and diabetes, should be strictly monitored for the possibility of increased toxicities [180,216,217]. Furthermore, a recent retrospective study showed that treatment with Nivolumab plus Relatlimab could have a reduced efficacy in patients with type 2 diabetes, possibly due to a low expression of LAG-3 in tumor tissue [218].

For the described data, in patients with pre-existing conditions, ICIs should be prescribed after an accurate risk-benefit evaluation since patients with comorbidities could be exposed to a higher risk of AEs. However, to date, no pre-existing comorbidities represent an absolute contraindication to treatment with ICIs. More prospective studies are necessary to better understand the safety and efficacy profile of these patients.

### 4.7. Sex

The interplay between sex and the immune system is becoming more acknowledged and may represent an interesting perspective to explain results observed in the clinical setting. Females can mount a more effective immune response: they have more efficient macrophages and neutrophils [219], antigen presentation [220], as well as higher CD4+ T cells [221,222], whereas males present more abundant natural killer cells [222], CD8+ T cells [221], T regulatory cells [223], and preferential T-helper-17 skewing [224]. These data reflect a complex interplay between hormones, genetics, behavior, and microbiome [225,226,227]. Also, melanoma in men is frequently associated with higher tumor mutational burden [228], and this could be in part explained with a different ultraviolet light exposition.

These preclinical findings have led to generating the hypothesis of a higher benefit of ICIs in men. Several studies supported these hypotheses in various cancer types, including melanoma. In 2018, a large systematic review and meta-analysis including 20 randomized controlled trials and 11,351 patients (3632 with melanoma) found that the magnitude of benefit with ICIs was higher in men (HR 0.72) than in women (HR 0.86). The difference was confirmed also in the melanoma subgroup (HR 0.79 vs. 0.66) [229]. However, an updated meta-analysis published by another group in 2020 that included 13,721 patients from 23 trials disproved these findings with no differential benefit between men and women (HR 0.75 vs. 0.77 in the overall population; 0.68 vs. 0.83 in the melanoma subgroup) [230]. However, it could be difficult to address this debate by meta-analysis alone without information on patient-specific data.

A retrospective study analyzed 3985 patients based on the Dutch registry: men and women experienced similar ORR when treated with ICI monotherapy, but men had lower ORR compared to women following combination treatment with anti-CTLA-4 + anti-PD-1 (51% vs. 67%) [231]. Another population study based on the Surveillance, Epidemiology, and End Results (SEER)–Medicare-linked data showed that women had a 2-fold increased hazard of mortality compared with men among patients receiving Nivolumab plus Ipilimumab, while there was no difference in the anti-PD-1 cohorts [232]. Kudura et al. found in 103 melanoma patients a better response to immunotherapy in men, and this difference was associated with higher immune activation parameters in peripheral blood [233].

In conclusion, sex differences could play a role in differential responses to ICIs, but data are controversial. A better understanding of the implication of sex in immunity and treatment could provide new biological insights and therapeutic targets to offer a tailored treatment.

## 5. Conclusions

The identification of predictive factors of response to ICIs is a research topic in fast evolution in metastatic melanoma. To date, only LDH, CRP, neutrophil-to-lymphocyte ratio, lymphocyte-to-monocyte ratio, and the presence of hepatic metastases can be easily used as predictors of outcome in routine daily practice. However, shortly, many new biomarkers will help to define the best treatment strategy. The characterization of the gut microbiome combined with genomics (TMB and driver mutations evaluation), transcriptomics (INF-gamma signature and single cell analysis of T cells subsets), and immunological (tumor HLA expression level) features, as well as ctDNA and radiomic, represent the mostly investigated areas and will be soon useful in this setting. Only a holistic approach will provide the missing pieces of the puzzle to predict immunotherapies outcomes. The high cost of these approaches and the need for reproducibility are the main issues that clinicians are going to face in the next years. Also, the tissue availability could limit the possibility of performing multiple tests, especially when the tumor is inaccessible or tissue biopsy yields insufficient tumor content. For these reasons, future clinical trials should address which combination of predictors is the most accurate and cost-effective to reach a personalized approach in melanoma patients (Figure 2). We should also notice that most of the described factors are important to determine outcomes irrespective of treatment, but only a few could guide treatment choice. In this scenario, the evaluation of IFN-gamma GEP, PD-L1 expression, tumor HLA expression, and the on-treatment monitoring of ctDNA levels seems to be the most promising to guide escalating or de-escalating strategies permitting a tailored immunotherapy treatment.

## Figures and Tables

**Figure 1 cancers-16-00101-f001:**
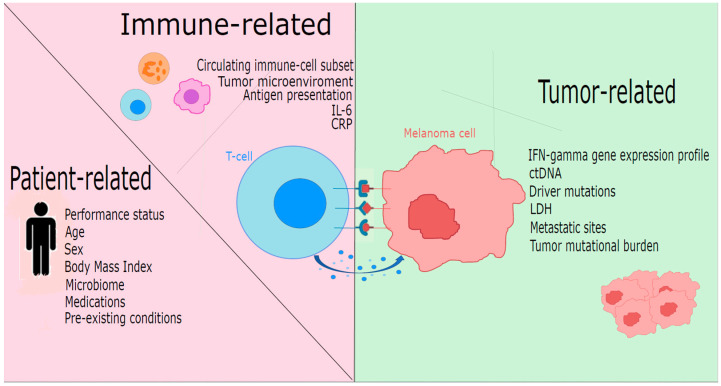
Predictors of response to immunotherapy in melanoma stratified by patient-related, immune-related, and tumor-related factors. ctDNA: circulating tumor DNA; CRP: C-reactive protein; IFN: interferon; IL6: interleukin 6; LDH: lactate dehydrogenase.

**Figure 2 cancers-16-00101-f002:**
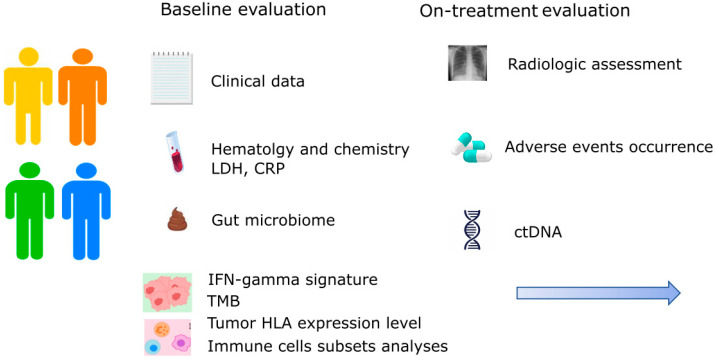
Future perspective in the application of multiple predictive factors of response to immunotherapy in metastatic melanoma. CRP: C-reactive protein; ctDNA: circulating tumor DNA; HLA: human leucocyte antigen; IFN: interferon; LDH: lactate dehydrogenase; TMB: tumor mutational burden.

**Table 1 cancers-16-00101-t001:** First-line predictive/prognostic factors. CRP: C-reactive protein; ctDNA: circulating tumor DNA; GEP: gene expression profile; HLA: Human leukocyte antigen; ICIs: immune checkpoint inhibitors; IFN: interferon; LDH: lactate dehydrogenase.

Factor	Source	PROs	CONs	Main Utility
Antigen presentation [12,13,14,15,16,17,18,19,20,21,22]	Tumor sample	High HLA class I expression correlated with better outcomes. Loss or downregulation in HLA class I molecules is a mechanism of resistance.	Method for HLA evaluation not yet established. Role of HLA class II is less established. Role of antigen-presenting machinery needs further evaluation.	Possible evaluation of different strategies in patients with established mechanisms of resistance to ICI.
C-reactive protein [23,24,25,26]	Serum	Correlation with tumor burden and worse outcomes.	Elevated CRP does not preclude response to ICIs.	Prognostic factor.
ctDNA [27,28,29,30,31,32,33,34]	Peripheral blood	Surrogate biomarker for tumor burden. High ctDNA levels correlate with worse outcomes.	25% of cases do not have a clear mutational driver to be monitored.	Useful to identify patients with worse prognoses and to anticipate progressive disease. Role as a non-invasive dynamic biomarker to anticipate radiologic progression.
IFN-gamma gene expression profiles [16,35,36,37,38,39,40,41,42,43]	Tumor biopsy	Highly predictive of inflammatory microenvironment. Role confirmed also in the neoadjuvant setting.	Absence of a unique validated signature. High costs per patient. Necessary but not sufficient to predict response to ICIs.	Possible escalating or de-escalating strategies according to low or high IFN-gamma GEP.
LDH [44,45,46,47,48,49]	Serum	LDH isoenzymes determine immunosuppressive microenvironment.	Elevated LDH does not exclude response to ICIs, especially in combination therapies.	Well-established prognostic factor.
Metastatic sites [44,50,51,52,53,54]	Radiologic assessment	Patients with hepatic metastases showed worse prognosis.	Fewer data on other metastatic sites. Lack of data to associate tumor burden and response to ICIs.	Patients with brain metastasis should be treated with ICI combination instead of monotherapy.
Tumor mutational burden [16,40,55,56,57,58,59,60]	Tumor sample	Synergic role with IFN-gamma GEP.	Confounding factors such as melanoma subtype could lead to misinterpretation. Determination methods are not standardized.	Promising predictive factor of response to immunotherapy, especially in combination with IFN-GEP.

**Table 2 cancers-16-00101-t002:** Second-line predictive/prognostic factors. ICIs: immune checkpoint inhibitors; IL: interleukin; MLR: monocyte-to-lymphocyte ratio; NLR: neutrophil-to-lymphocyte ratio; PLR: platelets-to-lymphocyte ratio; TILs: tumor-infiltrating lymphocytes.

Factor	Source	PROs	CONs	Main Utility
Checkpoint immunohistochemical evaluation [4,6,7,61,62,63,64,65,66]	Tumor sample	Possible use to select monotherapy or combination.	Inducible and heterogeneous biomarkers. Inconclusive relationship with prognosis.	Patients with PD-L1 negative tumors could benefit more from ICI combinations than monotherapy
Gut microbiome [67,68,69,70,71,72,73,74,75,76,77,78,79,80]	Stool sample	Gut microbiome composition associated with response to ICIs. Better predict unfavorable outcomes.	Mechanism not fully understood. Methodic validation is needed.	Potential therapeutic implication (e.g., diet intervention, fecal transplant).
IL-6 [25,26,81]	Serum	High level is associated with poor outcomes.	Not confirmed in large trial.	Potential prognostic factor.
Immune-related adverse events (irAEs) [6,82,83,84,85,86,87,88,89]	Medical history and clinical examination	Vitiligo is associated with better outcomes.	Less clear association with other toxicities.	Vitiligo could be an epiphenomenon of disease response.
Radiomic [90,91,92,93,94,95,96,97,98,99,100,101,102]	Radiologic assessment	Radiomic features reflect tumor biology of the entire tumor burden.	Time necessary for the manual segmentation of tumor lesions limits its clinical application. Limited reproducibility between different centers.	Predictive/prognostic features need to be further clarified.
Tumor microenvironment and peripheral blood cell subsets evaluation [52,103,104,105,106,107,108,109,110,111,112,113,114,115,116,117]	Peripheral blood/ Tumor sample	Higher TILs infiltration correlates with better outcomes. Presence of myeloid cells associated with resistance to ICIs. High NLR, MLR, and PLR are associated with poor	Different subpopulations are associated with outcomes. Absence of comparative data.	Higher potentiality for single-cell and topographical evaluation.

**Table 3 cancers-16-00101-t003:** Third-line predictive/prognostic factors. AID: autoimmune disease; BMI: body mass index; ICIs: immune checkpoint inhibitors; irAEs: immune-related adverse events; PS: performance status.

Factor	Source	PROs	CONs	Potential Utility
Age [118,119,120,121,122,123,124,125,126,127,128,129,130,131,132,133,134,135,136]	Medical history and clinical examination	Higher risk of discontinuation in elderly.	Similar efficacy in young and old patients.	Elderly patients should be monitored due to a possible higher impact of irAEs.
Body mass index (BMI) [137,138,139,140,141,142,143,144,145,146,147,148,149,150,151,152,153]	Medical history and clinical examination	Possible better outcomes in obese patients.	Controversial data.	Different features of body mass composition could better perform.
Concomitant medications [154,155,156,157,158,159,160,161,162,163,164,165,166,167,168,169,170,171,172,173,174,175,176,177,178,179,180]	Medical history and clinical examination	Steroid and antibiotic use is associated with worse outcomes.	Debate if causative or associative relationship. Absence of confirmed data for other medications. Only retrospective data.	Steroids and antibiotics should be prescribed with caution (except for irAEs treatment).
Driver mutations [6,181,182,183,184,185,186,187,188,189,190]	Tumor sample	Possible greater effect for ICI combinations vs. monotherapy in BRAF mutant.	NRAS negative impact not confirmed. ICIs are also effective in BRAF-mutant melanoma.	NRAS/BRAF status could not be used to select different ICI regimens.
Performance status (PS) [52,191,192,193,194,195,196]	Medical history and clinical examination	Poor PS correlates with worse outcomes.	Patients with poor PS could benefit from ICIs.	Prognostic factor.
Pre-existing conditions [180,197,198,199,200,201,202,203,204,205,206,207,208,209,210,211,212,213,214,215,216,217,218]	Medical history and clinical examination	Risk to exacerbate the pre-existing AID during ICIs.	No comorbidities represent an absolute contraindication to ICIs.	Patients should be monitored for an increased risk of adverse events.
Sex [219,220,221,222,223,224,225,226,227,228,229,230,231,232,233]	Medical history and clinical examination	Possible higher benefit with ICIs in men.	Controversial data.	Potential impact of sex on ICI response should be further explored.

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
