# Peer review of "Predictive Factors in Metastatic Melanoma Treated with Immune Checkpoint Inhibitors: From Clinical Practice to Future Perspective"

_cancers, 2023, doi:10.3390/cancers16010101_

Round 1

Reviewer 1 Report (Previous Reviewer 2)

Comments and Suggestions for Authors

In this new revision, the authors have failed to address the main question: which of the 23 factors/biomarkers listed in Table 1 are critical, from the clinical perspective, i.e. which ones carry the most weight, potentially affecting clinical decision-making, and which ones are less important. Is TMB less important than "myeloid cells", and CRP more important than ctDNA?

While individual sections pertaining to the 23 factors are well reviewed, the main structure is lacking. Simply put, the long list of 23 factors is not useful in its current form. Furthermore, some of the factors will affect treatment choice (e.g., aPD1 versus combination IPI+Nivo), and/or sequence.

Whether the factors are predictive and/or prognostic, should also be indicated early on and not in the fig 2 summary at the end of the review.

In terms of Fig 1, the distinction between immune related and tumour related factors appears arbitrary. T cell subsets (immune) versus TILs (tumour) – what is the difference? IFNg signatures are typically tumor-based and assessed on-therapy; why is CRP tumor related and IL-6 immune related, and why are myeloid cells not immune related? Adverse events are not “predictive” (by definition, they do not precede treatment); complete blood count as per table 1 is no different from "myeloid cells" as myeloid ratios are derived here.

Minor.

Table 1, Vitiligo was correct, while “cutaneous toxicity” (changed to) is not

Table 1, TILs are mentioned twice

P2 line 51, OS already used above; line 77, "acquired resistance" is a more commonly used term

Comments on the Quality of English Language

The English language is fine, minor editing is required

Author Response

Reviewer 2 Report (Previous Reviewer 3)

Comments and Suggestions for Authors

Dear authors,

All the comments suggested are resolved.

Best regards. 

Author Response

Thank you very much for your support and appreciation.

Round 2

Reviewer 1 Report (Previous Reviewer 2)

Comments and Suggestions for Authors

I would like to congratulate the authors on the new version of the manuscript. The structure is much clearer, and the manuscript is infinitely more readable. I have a few suggestions that I hope the authors will take on board, and several minor issues that need to be fixed before acceptance.

1.     Sections 2, 3, 4 and the corresponding tables. I suggest using “first line predictive/prognostic factors”, “second line predictive/prognostic factors” and “third line predictive/prognostic factors” as headings/titles.

2.     Citations need re-formatting and checking as there is a misalignment in referencing starting from ~#70 (all numbers shifted by one, thus referencing largely incorrect).

3.     Conclusions are to be refined with respect to two points: discussion with regards to the level of predictive accuracy and the obstacles; and the value of predictive factors in determining treatment choice versus determining treatment outcomes.

Other points.

·      There are several occasions when the authors use singular nouns instead of plural e.g. line 36, 524 (subsets not subset) and multiple others.

·      Line 50, (CTLA-4)  Ipilimumab – add “antibody” before IPI

·      Table 2 CRP: preclude not exclude

·      Table 2 ctDNA: remove “radiologic” as doesn’t fit context; move “role as non-invasive biomarker…” to the last column

·      Table 2 TMB, define “mutational assets” or rephrase

·      Line 176, define ECOG

·      Section 2.4, add “IFNg” before GEP

·      Line 198, “genomic sequences” – GEP are transcriptomics not genomics, please correct here and later. Also applies to section 5 Conclusions. Please check any other mentions.

·      Line 224, I wonder if the authors would like to discuss tumors with intrinsic IFNg signaling here.

·      Table 3, I assume Gut microbiome here

·      Line 330, “ligands” is more accurate than “molecules” in this context

·      Line 335, a sentence linking PDL1 expression to IFNg, is needed

·      Line 354, LAG3 is in an inhibitory checkpoint not a checkpoint inhibitor

·      Line 357, “ligands” should be plural

·      Line 374, change “its level” to “IL6 level”

·      Section 3.3. Define what kind of microbiome

·      Line 440, I assume what the authors mean here, is organ metastases ratio not organ size ratio? If mets, would these be defined as organ presence, number or total burden per organ?

·      Line 502, “lymphocytes” should be plural

·      Line 533, CD163+ add “macrophages”

·      Table 3, Vitiligo is strongly associated with better outcomes and perhaps the authors should consider moving into another table

·      Line 569, should be “a trend favoring older patients”

·      Line 583, Obesity and gut microbiome – use “link” or similar, not “influence” as causation is likely more complex than implied

·      Line 586, edit sentence e.g. “six cohorts with a total of x metastatic melanoma patients”, or similar

·      Line 594, use either “other” or “successive”, not both

·      Line725, a typo - should be “adverse”

·      Line 772, I assume retrospective analyses 

·      Line 820, define relatlimab

·      Lines 831 and 832, Females and males should be plural

·      Lien 834, how is microbiome defined by sex?

·      Line 866, what is PCR here? a typo for CRP or BRAF status evaluation?

·      Line 871, “HLA evaluation” would normally mean HLA typing - do the authors mean tumor HLA expression level, or presence of certain HLA alleles in a patient?

·      Line 870, IFNg expression should be moved to transcriptomics; genomics should include driver mutations – BRAF/NRAS status in the very least. I would also disagree with the authors as to the definition of routine tests as many are already available as routine path tests today.

Comments on the Quality of English Language

Moderate English editing is required. The manuscript would benefit greatly from professional editing.

Author Response

This manuscript is a resubmission of an earlier submission. The following is a list of the peer review reports and author responses from that submission.

Round 1

Reviewer 1 Report

Comments and Suggestions for Authors

·         Brief information about evolution of immunotherapy related to metastatic melanoma treated with immune checkpoint inhibitors is required.

·         Authors should consider the effect of pre-existing diseases of host that could influence response to ICIs.

·         Elaboration of the figure legends with details (figure 1) should be done.

·         It is well known that treatment with ICI frequently leads to severe immune-related adverse events (irAE). Authors should consider and discuss that.

·         There is no information about advantages of multiple predictive biomarkers to ICIs efficacy.

Comments on the Quality of English Language

Minor editing of English language required

Reviewer 2 Report

Comments and Suggestions for Authors

In the manuscript entitled “Predictive factors in metastatic melanoma treated with immune checkpoint inhibitors: from clinical practice to future perspective”, Poletto et al review predictive biomarkers of immunotherapy response in advanced melanoma. The authors group predictive biomarkers into 3 groups: patient-related, immune-related and tumor-related, and then analyze each category in more detail. The authors conclude that many of the biomarkers they have described, were actually not predictive of immunotherapy response, some were, and for many the association is unclear. No hierarchy, or any useful grouping, is provided; although the authors attempt at pointing out the “better” predictive biomarkers, these get buried among the less useful/irrelevant ones.

The topic is popular, and multiple reviews have been published in the last couple of years, and continue to be published; this paper does not make a new contribution to the topic – it is poorly written, convoluted, often uses unsubstantiated statements (i.e. not properly referenced), often cites other reviews rather than original studies -  but my main objection is that there is no clearly identified, core set of best predictive response biomarkers, with some estimate of their specificity and sensitivity.

Specific points:

When using terms like “predictor of survival”, “predictor of outcomes”, the authors need to make it clear whether there was a positive or a negative association; “trend” is not a [statistically significant] difference, and as such meaningless

Figure 1: Specify T cell subsets; specify PCR

Table 1:  Better separation of sections required

Line 133: “Women can mount a more effective immune response: they have more efficient macrophages and neutrophils, antigen presentation, and B cell response, as well as …” – there is no basis for this claim. There are other similarly unsubstantiated claims throughout.

Section 4.4 HLA – there is a different association with MHC class I and MHC class II expression, and ICI response in melanoma

Line 485, PD1 + T cells not PD-L1+ in this study

Section 5.4 ctDNA can be both predictive and prognostic; it is not just about tumor burden but also about tumor evolution

Line 518, “Even if the association between low plasma ctDNA and longer PFS may be obvious, it may be more inquisitive in the field of immunotherapy if we consider the possible lower antigen exposition in patients with lower tumour burden” – what does this mean?

Line 553, “Management of patients with early SD is an important unmet need” – what do the authors mean? Do they suggest a change in treatment for this group?

Section 5.6 – “Mutational asset” – driver mutations?

Comments on the Quality of English Language

Poor - extensive editing required prior to submission. Also applies to sentence structure, terms usage, tense usage; there are multiple typos throughout

Reviewer 3 Report

Comments and Suggestions for Authors

I would like to thank the authors for the opportunity to review their paper. It is an excellent review about the most aggressive skin cancer: melanoma. In the last years, the immunotherapy and target therapy are more frequent used to treat metastatic melanoma, therefore it is important to appreciate, more accurate their response.  

Hereby please find my comments regarding the paper:

- Please verify that all abbreviations are explained in the text (some are missing).

- Some editorial error exists in the text (lines 32, 475, 636).

- In line 163-167 I change the terms sex with gender.

Round 2

Reviewer 2 Report

Comments and Suggestions for Authors

Poletto et al present an improved version of the manuscript, however the main question remains:

The authors stipulate that they wish to help clinicians support “daily practice decisions”, yet they have come up with the long list of 23 predictive biomarkers and factors that may affect the ICI response; which of these factors/biomarkers are critical, from the clinical perspective, and can affect clinical decision-making? The authors make an attempt to solve this in Fig 2 but this does not go far enough to be a useful guide.

Could the authors provide a brief list of PROs and CONs for each parameter in Fig 2?

As a note, IFNg signature and T cell subset analysis should also be included as on-treatment evaluation tests (change from the baseline being the best prediction of response to date)

Minor comments

Line 65, “In patients that harbored BRAF mutation…” – Tumors have BRAF mutations not patients

In Line 85, authors mention, “Moreover, biomarkers could help to assess the decision between the different ICIs combinations available and to define patients that could benefit from a from a triplet combination therapy with BRAF inhibitors, MEK inhibitors, and anti-PD-(L)1” – How exactly?

Line 99, “we summarize the most important factors supposed to be associated..” – they are, or they are not.

Comments on the Quality of English Language

English is fine, minor editing only